# Image Super-Resolution with Guarantees via Conformalized Generative Models

**Eduardo Adame**
School of Applied Mathematics
Getulio Vargas Foundation
eduardo.salles@fgv.br

**Daniel Csillag**
School of Applied Mathematics
Getulio Vargas Foundation
daniel.csillag@fgv.br

**Guilherme Tegoni Goedert**
School of Applied Mathematics
Getulio Vargas Foundation
guilherme.goedert@fgv.br

## Abstract

The increasing use of generative ML foundation models for image restoration tasks such as super-resolution calls for robust and interpretable uncertainty quantification methods. We address this need by presenting a novel approach based on conformal prediction techniques to create a 'confidence mask' capable of reliably and intuitively communicating where the generated image can be trusted. Our method is adaptable to any black-box generative model, including those locked behind an opaque API, requires only easily attainable data for calibration, and is highly customizable via the choice of a local image similarity metric. We prove strong theoretical guarantees for our method that span fidelity error control (according to our local image similarity metric), reconstruction quality, and robustness in the face of data leakage. Finally, we empirically evaluate these results and establish our method's solid performance.

## 1 Introduction

Generative ML foundation models led to massive leaps in the capabilities of modern image synthesis and processing, spanning domains such as image generation, inpainting, and super-resolution. Particularly in the case of image super-resolution, recent methods have become considerably adept at leveraging patterns in images to better recover complex textures, geometries, lighting and more.

In testament to these improvements, leading manufacturers are constantly improving and deploying tools based on these frameworks in every new generation of consumer devices. These widespread applications highlight an important question: How trustworthy are the predictions of these models? When a model does some particular inpainting or super-resolution infill, what guarantees do we have that its predictions are truly accurate to reality, and not mere hallucinations? Therefore, it would be most desirable to have a proper uncertainty quantification over the predicted image.

However, most of the previous contributions to this endeavor have suffered from a lack of interpretability. In order to be widely adopted, a new framework should clearly communicate its uncertainty estimates to the public in a way that reflects how they will be used. This demand for interpretability fundamentally guides the properties and theoretical guarantees we seek to establish for our predictions, and thus on the underlying procedure. Of course, all of this is compounded by the usual challenges of having to do trustworthy uncertainty quantification that is model agnostic and can be employed effectively atop any "black-box" foundation model.

39th Conference on Neural Information Processing Systems (NeurIPS 2025).

In this paper, we address all these issues by proposing a method based on techniques from conformal prediction [Vovk et al., 2005] and conformal risk control [Angelopoulos et al., 2022a], while employing metrics designed for interpretability in concrete applications. All our method requires is a handful of unlabelled high-resolution images that were not present in the training set for the diffusion model, and we achieve strong guarantees on our predictions that are also intuitive to the user and robust to violations of our key assumptions.

Our main contributions are:

- A new method to quantify uncertainty in images inpainted or augmented by diffusion-based models. Our method can work atop any black-box diffusion model, including models that are locked behind an opaque API, and requires only easily-attainable unlabelled data for calibration.

- We identify additional theoretical guarantees enjoyed by our model. In particular, we prove that our method also controls the PSNR of the predicted images, and show that it is reasonably robust to data leakage, reinforcing the effectiveness and robustness of our method.

- A comprehensive study of modelling choices in our approach, revealing particular modifications from the base procedure that can significantly enhance performance. Particularly, we demonstrate that certain applications of low-pass filters can greatly improve our method's effectiveness.

We note that our procedure is also reasonably generalizable to other image restoration tasks, and provide examples in the supplementary material.

**Related work**   There have been some attempts on formal uncertainty quantification for image super-resolution with varying approaches. But most existing works, e.g., [Angelopoulos et al., 2022b, Teneggi et al., 2023], have as their goal to produce, rather than a single image, an interval-valued image (i.e., an image where each pixel is represented by an interval rather than a single value), and ensure that these intervals will, with high probability, contain the 'true' pixel values. However, this is fairly non-intuitive for the user and its underlying guarantees are a bit lax. As far as we are aware, the only existing solution that proposes some other way of quantifying uncertainty is [Kutiel et al., 2023], which produces a continuous "confidence" mask over the predicted image, meaning that each pixel in the image is assigned a confidence score in $[0, 1]$. However, their solution has no probabilistic guarantee, which is fundamental for reliable uncertainty quantification (see the supplementary material). Ours, in contrast, is backed by a plethora of such theorems. Finally, our proposed solution is closely related to the existing applications of conformal prediction to semantic segmentation, e.g., [Csillag et al., 2023, Mossina et al., 2024]. Also worth noting are methods for uncertainty quantification for image super-resolution that do not have any formal guarantee of correctness; relevant works include both general uncertainty quantification techniques for deep learning such as Bayesian neural networks [Goan and Fookes, 2020] and Monte Carlo Dropout [Gal and Ghahramani, 2015], as well as more task-specific approaches such as [Nehme et al., 2023, Adapa et al., 2024].

## 2   Method

### 2.1   Conformal Mask Calibration

Let us be supplied with any generative image super-resolution model $\mu : [0, 1]^{w \times h \times 3} \rightarrow [0, 1]^{kw \times kh \times 3}$, where $w$ and $h$ stand for the low-resolution image dimensions, and $k$ is the up-scaling factor. Naturally, this is a stochastic function due to the generative nature of the model, so an intuitive (albeit nonrigorous) way to quantify model indecision would be to aggregate many realizations of the output image (e.g. by computing the variance of generated pixels). However, our methodology is capable of working naturally when supplied with an arbitrary estimate of model indecision that can be described by a function $\sigma : [0, 1]^{w \times h \times 3} \rightarrow \mathbb{R}^{kw \times kh}$. We further discuss useful constructions of $\sigma$ in Section 2.2.

Having received a lower resolution image $X$, we must consider how the model prediction $\widehat{Y} = \mu(X)$ differs from the true high resolution image $Y$. Our goal is to find a "confidence mask" $M(X)$ that

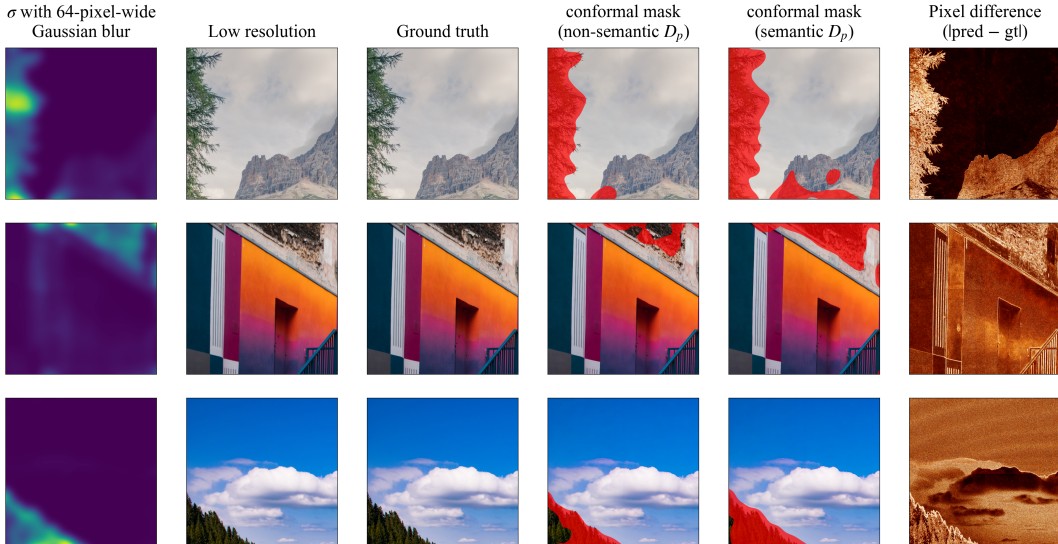

| $\sigma$ with 64-pixel-wide Gaussian blur | Low resolution | Ground truth | Prediction with conformal mask (non-semantic $D_p$) | Prediction with conformal mask (semantic $D_p$) | Pixel difference (\|pred − gt\|) |

Figure 1: **Our method highlights meaningful uncertainty regions in generated images.** This figure presents a comparison of multiple high-resolution images with their corresponding conformal masks. Our conformal masks accurately highlight regions where the predictions significantly deviate from the ground truth, capturing differences in color, texture, and lighting.

indicates the region of the image whose content we trust, i.e. the pixels in the predicted image with indecision below a sought threshold. Formally, the mask is a (possibly stochastic) function $M : [0,1]^{w \times h \times 3} \to \{0,1\}^{kw \times kh}$ that has image $M(X) = \{p \in \mathbb{Z} \times \mathbb{Z} : [\sigma(X)]_p \leq t\}$, where $[\bullet]_p$ is the image value at pixel $p$ (be it binary, grayscale or colored) and $t$ is a desired indecision threshold.

We seek masks that satisfy fidelity guarantees between the true and predicted high-resolution images. This fidelity is measured by a **fidelity error** defined as $\sup_{p \in M} D_p(Y, \widehat{Y})$, where $D_p$ is a function that measures the difference between two images around some pixel location $p$. We can employ any local measure of the difference as long as $0 \leq D_p(Y, \widehat{Y}) \leq 3$ for all $p, Y$ and $\widehat{Y}$ (e.g. $D_p(Y, \widehat{Y}) = \|[Y]_p - [\widehat{Y}]_p\|_1$). There are many different useful choices for $D_p$, with a few explored in Section 2.3.

Equipped with the previous definitions, we are able to produce these fidelity masks $M_\alpha(X)$ for any desired fidelity level $\alpha \in [0,1]$ with the guarantee that

$$\mathbb{E}\left[\sup_{p \in M_\alpha(X)} D_p(\mu(X), Y)\right] \leq \alpha$$

by thresholding the output of $\sigma$ by some parameter $t$ in the construction of the mask. This parameter can then be calibrated for by using techniques of conformal prediction [Vovk et al., 2005] and conformal risk control [Angelopoulos et al., 2022a] with just access to unlabelled hold-out data, which has not been used to obtain either $\mu$ or $\sigma$ (if there is data contamination, weaker guarantees hold; see Proposition 3.2). In particular, given such data $(X_i, Y_i)_{i=1}^n \subset [0,1]^{w \times h \times 3} \times [0,1]^{kw \times kh \times 3}$, we produce

$$t_\alpha = \sup\left\{t \in \mathbb{R} \cup \{+\infty\} : \frac{1}{n+1}\sum_{i=1}^n \sup_{p;[\sigma(X_i)]_p \leq t} D_p(Y_i, \mu(X_i)) + \frac{3}{n+1} \leq \alpha\right\}, \quad (1)$$

thus obtaining

$$M_\alpha(X) := \{p \in \mathbb{Z} \times \mathbb{Z} : [\sigma(X)]_p \leq t_\alpha\}. \quad (2)$$

Crucially, all of our guarantees will hold for any $\mu$ and $\sigma$, though the produced trust masks $M_\alpha$ will generally be larger (i.e., more confident) for well-chosen ones.

This methodology comes with a strong statistical assurance: a marginal "conformal" guarantee. It holds in expectation on the calibration data jointly with an additional new, 'test' sample:

**Theorem 2.1** (Marginal conformal fidelity guarantee). *Let $\alpha \in \mathbb{R}$ and $n \in \mathbb{N}$. Suppose we have $n + 1$ i.i.d.[1] samples $(X_i, Y_i)_{i=1}^{n+1}$ from an arbitrary probability distribution $P$ and let $t_\alpha$ and $M_\alpha$ be as in Equations 1 and 2 (and thus only a function of $X_i, Y_i$ with $i = 1, \ldots, n$). Then it holds that*

$$\mathbb{E}_{(X_i, Y_i)_{i=1}^{n+1}} \left[ \sup_{p \in M_\alpha(X_{n+1})} D_p\left(\mu(X_{n+1}), Y_{n+1}\right) \right] \leq \alpha.$$

Thanks to the discrete combinatorial structure of the set the infimum is taken over in Equation 1, the indecision threshold $t_\alpha$ (defined in Equation 1) can be efficiently computed with the use of dynamic programming in $O(nd \log d)$ time, where $n$ is the number of images in the calibration set and $d$ is the number of pixels in each image. In contrast, a naive brute force algorithm would take $\Omega(n^2 d^2)$ time. The relevant pseudocodes can be found in the supplementary material.

## 2.2 Producing Score Masks

A key component of our algorithm is the model indecision estimate $\sigma : [0,1]^{w \times h \times 3} \to \mathbb{R}^{kw \times kh}$. A good $\sigma$ should attain higher values for regions of the image where there is more uncertainty, and lower values for regions where we are more certain. Nevertheless, our guarantees hold for any choice of $\sigma$.

Considering the generative nature of our base models, one natural way to produce such a $\sigma$ is to take the pixel-wise empirical variance of $M$ generated images:

$$[\sigma^{\mathrm{var}}(X)]_p = \widehat{\mathrm{Var}}_M\big[[\mu(X)]_p\big].$$

However, this may suffer from being too local: for example, if the model correctly knows that an edge must be present in a particular region of an image but slightly misplaces it by one or two pixels, there would be a significant mismatch between the "true" model uncertainty and the indecision estimate by $\sigma^{\mathrm{var}}$.

To resolve this, we propose to 'smooth out' our predictions by performing a convolution with a low pass kernel $K$. A naive way of doing so would be to convolve the images whose pixels we are taking the variances of directly: $\widehat{\mathrm{Var}}_M\big[[\mu(X) * K]_p\big]$. However, this has an unintended side effect: by applying the convolution directly to the generated images, we are effectively undoing the super-resolution done by the model! Hence, we propose to instead apply the convolution to the computation of the variance, via its decomposition in terms of the second moment:

$$[\sigma^{\mathrm{ker}\text{-}K}(X)]_p = \widehat{\mathbb{E}}_M\big[[\mu(X)^2 * K]_p\big] - \left(\widehat{\mathbb{E}}_M\big[[\mu(X) * K]_p\big]\right)^2.$$

It should be noted that when $K$ is a 1-box kernel, we recover $\sigma^{\mathrm{var}}$. After computing this patch-based variance, we further apply a Gaussian blur to the resulting variance map. This additional smoothing step helps mitigate the risk of border artifacts being overly emphasized, which can otherwise lead to an undesired overestimation of uncertainty along edges.

Finally, we remark that ideally this model indecision would be estimated jointly with the upscaled image $\mu(X)$. This is, however, fairly nontrivial and best left for future work.

## 2.3 Choices of $D_p$

A crucial point of our procedure is the definition of the precise fidelity error we are controlling. This is given by the choice of $D_p$, which is a function indexed by a pixel position $p$ that receives the real and predicted images $Y$ and $\widehat{Y}$ and returns a notion of how similar the two are around $p$. Our procedure is valid for any choice of $D_p$ that is bounded within $[0,3]$, though it is best to chose one for which $D_p(Y, \widehat{Y}) \to 0$ as $\widehat{Y} \to Y$. Here we highlight a couple of the most natural and useful:

**Pointwise metric** $\quad D_p(Y, \widehat{Y}) = \|[Y]_p - [\widehat{Y}]_p\|_1$, where $[Y]_p$ and $[\widehat{Y}]_p$ correspond to the pixel color of $Y$ and $\widehat{Y}$ at pixel position $p$ (and thus the 1-norm is necessary to condense their difference into a single number).

---

[1]Technically, Theorem 2.1 holds under the weaker assumption of exchangeability, with the same proof. We stick to i.i.d. for simplicity.

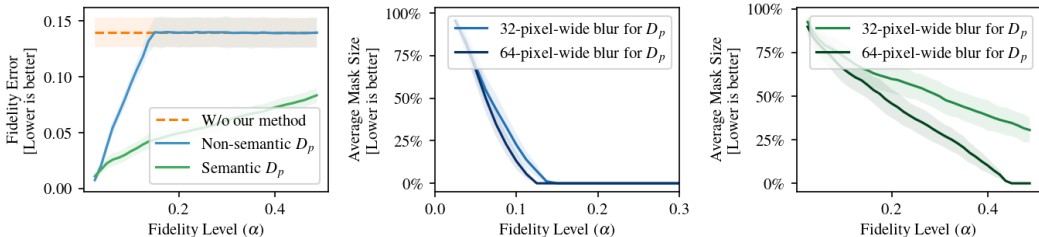

Figure 2: **Our method provably controls the fidelity error with accuracy.** *Left:* The figure shows the non-semantic fidelity error obtained by our method for varying fidelity levels $\alpha$, for calibration with both the semantic $D_p$ (orange) and non-semantic $D_p$ (blue). As is shown in our theoretical guarantee, the error is tightly controlled by our method, being at most $\alpha$. *Center and Right:* The plot displays the size of our confidence masks (i.e., how much of the image we do not trust) for varying fidelity levels $\alpha$, for calibration done with a semantic $D_p$ (*Center*) and a non-semantic $D_p$ (*Right*). As $\alpha$ increases our masks get smaller, eventually reaching zero, i.e., trusting the whole image.

**Neighborhood-averaged metric** $D_p(Y, \widehat{Y}) = \|[Y * K]_p - [\widehat{Y} * K]_p\|_1$, where $K$ is some convolution kernel corresponding to a low pass filter. This makes it so that single wrong pixels in the midst of many correct pixels do not influence the loss function too much, and generally leads to larger confidence masks.

In both cases, we consider the images in Lab color space. This ensures that all the color comparisons being done are perceptually uniform, which would not be the case in e.g., sRGB space.

Though perfectly valid and useful, it is well known that such pixel-wise comparisons struggle to capture semantic and perceptual properties of the underlying images. Hence, both of the options presented above struggle to truly capture semantic differences, where a user would clearly note a difference between the predicted and ground truth images. To this end, we propose a third option based on additional labelled data:

**Semantic metric** We can suppose a stochastic function $S : [0,1]^{kw \times kh \times 3} \times [0,1]^{kw \times kh \times 3} \rightarrow \{0,1\}^{kw \times kh \times 3}$ that indicates a mask produced by a human-being denoting the differences between the two given high-resolution images (a value of 1 on the image represents a differing point). We can then consider $D_p(Y, \widehat{Y}) = [S(Y, \widehat{Y})]_p$. Note that for the calibration procedure we only need to compute the $D_p$ on the calibration images, and thus the only samples of the human annotations $S(Y, \widehat{Y})$ that we need are on the calibration data.

## 3 Additional theorical results

In this section we present additional theoretical properties enjoyed by our method, which highlight its flexibility and robustness.

### 3.1 Our method provably controls PSNR

So far we have only proven results for the 'fidelity error' defined in Subsection 2.1. However, our results can also be directly mapped to more familiar metrics for image quality quantification. In particular, we can prove strong guarantees on the PSNR of our predictions, a common metric of image fidelity and quality in computer graphics:

**Proposition 3.1.** *Let $\alpha \in \mathbb{R}$ and $n \in \mathbb{N}$. Suppose we have $n + 1$ i.i.d. samples $(X_i, Y_i)_{i=1}^{n+1}$ from an arbitrary probability distribution $P$ and let $t_\alpha$ and $M_\alpha$ be as in Equations 1 and 2 (and thus only a function of $X_i, Y_i$ with $i = 1, \ldots, n$). Then it holds that*

$$\mathbb{E}_{(X_i,Y_i)_{i=1}^{n+1}} \left[ \text{PSNR}\left(\mu(X_{n+1}), Y_{n+1} | M_\alpha(X_{n+1})\right) \right] \geq -20 \log_{10} \alpha.$$

It is rather remarkable that, despite our procedure being originally designed in order to establish guarantees for uncertainty quantification, it also maps over to guarantees on a standard image quality

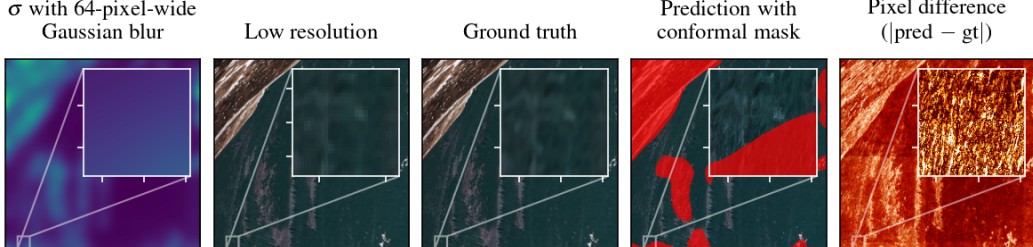

| σ with 64-pixel-wide Gaussian blur | Low resolution | Ground truth | Prediction with conformal mask | Pixel difference ($|\text{pred} - \text{gt}|$) |

Figure 3: **Unreliable predictions are accurately detected.** This figure shows a close-up example where $D_p$ is non-semantic (as shown in Figure 1), highlighting a failure of the base model to reconstruct a blurred area. Our method correctly identifies this failure and assigns low confidence to the affected region, where the predicted image deviates from the ground truth.

metric. The relative functional simplicity of the PSNR may be a contributing factor to this result, but we expect that similar finds will soon follow for other metrics (though the proof would be more involved; the full proof of Proposition 3.1 can be found in the supplementary material).

## 3.2 Under data leakage

One crucial assumption of Theorem 2.1 is that we assume that the calibration data is independent of the base diffusion model – i.e., that the calibration data is independent from (or at least exchangeable with) the data used to train the diffusion model.

Though arguably achievable through the collection of new data for calibration purposes, this is considerably harder to ensure when using foundation models which have been trained on massive datasets that attempt to span all data on the internet. Hence, it becomes essential to explore what happens when there is data leakage from the training data to the calibration data – i.e., some amount of data in the calibration samples is actually already present in the training data.

Proposition 3.2 provides worst-case bounds on the miscoverage error when there is data leakage. In particular, we consider that out of the $n$ calibration samples, $n_{\text{leaked}} < n$ are actually drawn from the training data (or some other arbitrarily different data distribution), while the remaining $n_{\text{new}} = n - n_{\text{leaked}}$ are truly independent of the training samples.

**Proposition 3.2.** *Let $\alpha \in \mathbb{R}$ and $n \in \mathbb{N}$, with $n = n_{\text{new}} + n_{\text{leaked}}$. Suppose we fit our procedure as per Equations 1 and 2 with $n$ data points. Out of these $n$ data points, suppose that the first $n_{\text{new}}$ are sampled from some arbitrary probability distribution $P$, and the latter samples (indexed by $n_{\text{new}} + 1, \ldots, n$) be sampled from some arbitrarily different probability distribution $Q$. Then, as we take a new sample $X_{n+1}, Y_{n+1}$ from distribution $P$, it holds that*

$$\mathbb{E}_{(X_i, Y_i)_{i=1}^{n+1}} \left[ \sup_{p \in M_\alpha(X_{n+1})} D_p \left( \mu(X_{n+1}), Y_{n+1} \right) \right] \le \alpha \cdot \frac{n_{\text{new}} + n_{\text{leaked}} + 1}{n_{\text{new}} + 1}.$$

Note that $Q$ could even be the empirical distribution of the data used to train the base generative model. This result shows that our conformal prediction scheme (and split conformal prediction in general) is somewhat robust to data leakage, as long as it is not too severe in relation to the amount of calibration samples.

## 4 Experiments

We dedicate this section to the empirical evaluation of our method, demonstrating its effectiveness.

**Data**   All experiments were conducted using the Liu4K dataset [Liu et al., 2020], which contains 1,600 high-resolution (4K) images in the training set and an additional 400 4K images in the validation set. The dataset features a diverse collection of real-world photographs, including scenic landscapes, architectural structures, food, and natural environments. We use the training set for calibration

| Fidelity Level | Semantic $D_p$ | | | Non-semantic $D_p$ | | |
|---|---|---|---|---|---|---|
| | Avg. PSNR | Avg. Fidelity Error | Avg. Conformal Mask Size | Avg. PSNR | Avg. Fidelity Error | Avg. Conformal Mask Size |
| $\alpha = 0.075$ | $32.75 \pm 1.55$ | $0.03 \pm 0.01$ | $0.77 \pm 0.07$ | $30.23 \pm 1.12$ | $0.07 \pm 0.01$ | $0.43 \pm 0.09$ |
| $\alpha = 0.100$ | $32.65 \pm 1.48$ | $0.03 \pm 0.01$ | $0.73 \pm 0.07$ | $28.64 \pm 0.93$ | $0.09 \pm 0.01$ | $0.23 \pm 0.06$ |
| $\alpha = 0.200$ | $31.63 \pm 1.32$ | $0.05 \pm 0.01$ | $0.60 \pm 0.08$ | $26.82 \pm 1.03$ | $0.14 \pm 0.01$ | $0.00 \pm 0.00$ |
| $\alpha = 0.300$ | $30.86 \pm 1.29$ | $0.06 \pm 0.01$ | $0.50 \pm 0.08$ | $26.82 \pm 1.09$ | $0.14 \pm 0.01$ | $0.00 \pm 0.00$ |
| w/o our method | $26.83 \pm 1.06$ | $0.14 \pm 0.01$ | N/A | $26.82 \pm 1.08$ | $0.14 \pm 0.01$ | N/A |

Table 1: **Quantitative evaluation of our method under semantic and non-semantic settings.** We evaluate the average PSNR, fidelity error, and average conformal mask size across different fidelity levels ($\alpha$) for both semantic and non-semantic $D_p$. For reference, we also include a baseline without our method (i.e., trusting the whole image). Overall our method tightly controls the fidelity error and PSNR while producing precise and informative masks.

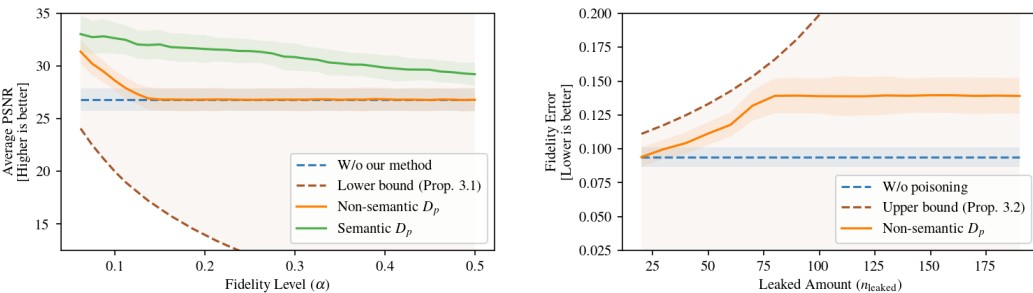

Figure 4: **Our controls the PSNR and is robust and under data leakage.** *Left:* This experiment confirms that the PSNR is theoretically bounded, as established in Proposition 3.1. Additionally, the PSNR within the conformal masks—in both semantic and non-semantic settings—remains consistently higher than the baseline, indicating improved prediction quality in trusted regions. *Right:* This plot illustrates that the fidelity error can bounded under data leakage, as per Proposition 3.2. In both plots, the values plateau once the method reaches the point of trusting the whole images.

procedures, and the test set for evaluation and metrics. We always work in Lab color space for its perceptual uniformity.

**Base Model**   We perform our evaluations atop SinSR [Wang et al., 2023], a state-of-the-art generative super-resolution method based on diffusion models. It performs super-resolution by conditioning the score function on a low-resolution image and applying diffusion in the latent space.

**Compute**   Experiments were run on an Intel Xeon E5-2696 v2 processor (2.5GHz base, 3.6GHz boost, 18 threads available) with 40GB of RAM and an NVIDIA RTX 6000 Ada Generation 48GB GPU. Notably, the primary computational bottleneck is the inference process of the base diffusion models, while the conformal calibration step is highly efficient and runs fairly quickly on a CPU. For reproducibility, the source code is available in `https://github.com/adamesalles/experiments-conformal-superres`, as well as in the supplementary material.

**Baselines**   We compare our method to the following prior work:

- **No uncertainty quantification:** this correponds to trusting the whole generated image. In terms of confidence masks, it is equivalent to producing an empty mistrust mask.

- **[Angelopoulos et al., 2022b]:** the output of this method is an image with interval-valued pixels, with a guarantee that, with high probability, the expected value of the percent of pixel intervals containing their true values will be at least $(1 - \alpha)$.

- **[Kutiel et al., 2023]:** the output of this method is a continuous score for each pixel (which they call a 'mask,' though their use of this term differs from ours), which is higher for regions of the image we can trust more. The idea is that when reweighting the pixels by

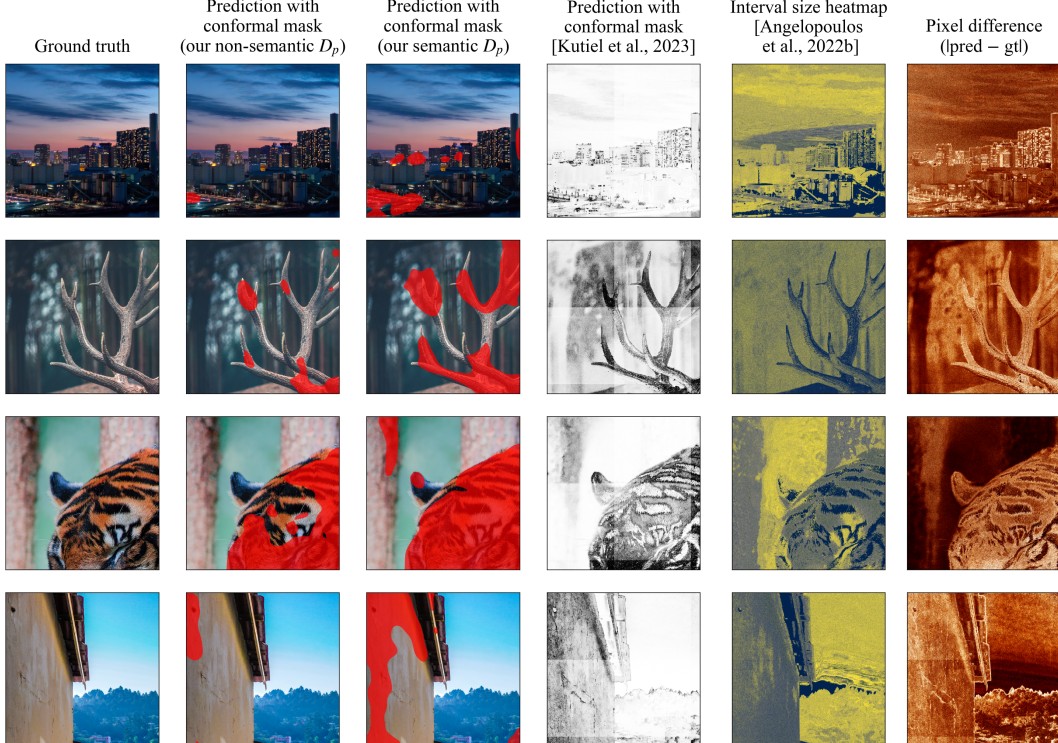

| Ground truth | Prediction with conformal mask (our non-semantic $D_p$) | Prediction with conformal mask (our semantic $D_p$) | Prediction with conformal mask [Kutiel et al., 2023] | Interval size heatmap [Angelopoulos et al., 2022b] | Pixel difference (|pred − gtl) |

Figure 5: **Qualitative comparison to prior work.** This figure compares our method – both semantic and non-semantic $D_p$, under the same settings as Figure 1 – against the methods of [Angelopoulos et al., 2022b] and [Kutiel et al., 2023]. While our conformal masks highlight precise regions of uncertainty in an interpretable way, the method from [Kutiel et al., 2023] produces continuous masks that closely mirror the original image rather than doing proper uncertainty estimation. Similarly, the heatmaps from [Angelopoulos et al., 2022b] do not visually convey uncertainty, making their interpretation more challenging.

this predicted score, the L1 loss of the generated image against the predicted image is, with high probability, at most some $\alpha$ in expectation. However, they do not prove such a validity property, and indeed it generally need not hold (see the supplementary material).

The prior work of [Angelopoulos et al., 2022b] and [Kutiel et al., 2023] produce pixel-wise intervals and scores, respectively, and are thus not directly comparable to our binary mask-based uncertainty quantification methodology. For this reason, on these two methods we are limited to a qualitative evaluation, which we do in Figure 5. That aside, we explore our method and show that it is a strict improvement over doing no uncertainty quantification.

In Figure 2, we analyze how well our procedure controls the fidelity error in practice, and how big the confidence masks get. We note that our theorems translate to excellent empirical performance, with the actual fidelity error closely following the specified fidelity level $\alpha$. Indeed, the two are essentially equal up until $\alpha$ becomes so large that we trust the whole image, at which point the fidelity error stays fixed. As for the mask sizes, they steadily decrease as $\alpha$ grows, remaining informative for all but the most extreme levels of $\alpha$. Table 1 reveals similar patterns.

As observed in Figure 3, our method successfully generates accurate confidence masks even when the base model fails in the super-resolution task. This applies not just to perceptual attributes such as color and brightness but also to cases where the original image is blurred; in such scenarios, the base model often misinterprets the blur as a loss of a higher frequency, and hallucinates. Fortunately, our method effectively captures these nuances, preserving essential details and producing faithful, trustworthy results.

In Section 3 we've theoretically shown that our method provably controls PSNR and has a certain robustness to data leakage; we now seek to empirically verify this. Figure 4(left) shows the PSNR

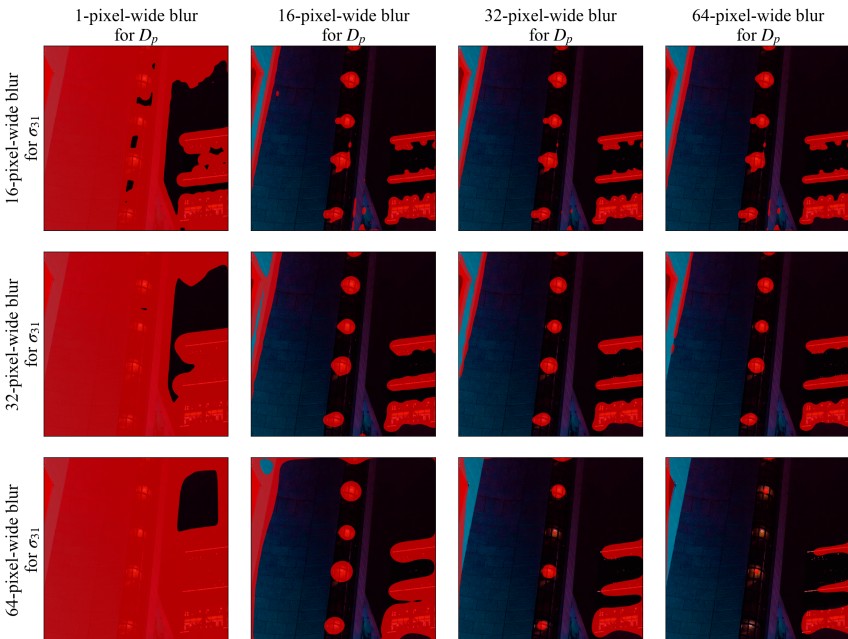

Figure 6: **Blurring improves our conformal masks.** Red regions correspond to the conformal confidence masks, denoting where we mistrust our predictions. Note how, as the radius of the Gaussian blur for $D_p$ increases, so does the coverage of the trusted regions; and, as the radius of the Gaussian blur for $\sigma$ increases, not only does the trusted cover also increases, but its regions become more contiguous and visually appealing.

obtained by our method (i.e., the PSNR of the reconstruction constrained to the region of the image we trust) over varying fidelity levels, along with the PSNR obtained by the base reconstruction and our bound from Proposition 3.1. We see that our bound is somewhat lax – especially for higher levels of $\alpha$ – but it is valid. Similarly, Figure 4(right) seeks to empirically measure our method's robustness to data leakage. For each level of data leakage ($n_{\text{leaked}}$) we run our method's calibration with $n_{\text{leaked}}$ of our samples being "perfect predictions" where $D_p \equiv 0$, i.e., samples that we are perfectly overfit on. Again we see the validity of our bound, and note that it is more precise for smaller amounts of leakage. After a certain point the fidelity error stabilizes at the highest possible fidelity risk (i.e., the fidelity risk when we trust the whole image).

We also investigate the size of the predicted confidence masks, a crucial aspect in assessing the reliability of our method. Notably, the choice of $D_p$ plays a fundamental role in determining how much confidence we can assign to our predictions. If $D_p$ is too local (e.g., defined simply as $D_p(Y, \widehat{Y}) = \|[Y]_p - [\widehat{Y}]_p\|_1$), the model lacks sufficient spatial context to make robust assessments, resulting in overly conservative confidence estimates and smaller masks.

By incorporating additional contextual information into $D_p$, such as smoothing the pixel-wise differences with a low-pass filter, we observe a substantial improvement in mask size and quality. This approach allows for larger, more informative confidence masks, as it accounts for a broader spatial region rather than relying solely on isolated pixel differences. As a result, the predictions not only become more faithful to the underlying image structure but also exhibit greater visual coherence. Empirically, this effect is evident in Figure 6, where the use of a smoothed $D_p$ leads to masks that better capture the regions of uncertainty while preserving the overall perceptual integrity of the image. These findings reinforce the importance of designing $D_p$ to effectively balance local accuracy with global consistency, ultimately enhancing both interpretability and reliability in uncertainty estimation.

Finally, we compare against all of our baselines in Figure 5. Our outputs are much more immediately interpretable and useful than those of [Angelopoulos et al., 2022b], whose interpretation is tricky due to the combination of a lower+upper image. Similarly, note that though the pixel scores of [Kutiel et al., 2023] are comparatively more interpretable, they are less informative as they follow the original image too closely, whereas ours selects meaningful regions of the image.

# 5    Conclusion

In this work, we have presented a new method for performing uncertainty quantification for image super-resolution based on generative foundation models endowed with statistical guarantees. Our method requires only easily attainable unlabeled data and is adaptable over any base generative model, including those locked behind an opaque API. We also prove that our proposed solution satisfies properties beyond that of conformal risk control, further strengthening it. We expect our method to be broadly useful in practice, including in scenarios beyond image super-resolution.

Nevertheless, our method has a few key limitations: for one, we assume that our calibration data together with a sample from the test set are exchangeable; effectively, this corresponds to giving guarantees for in-distribution behaviour. Though conformal prediction methods are known to be relatively robust to some changes in distribution, significant distribution shift may void our guarantees in principle. Moreover, it is worth noting that our algorithm requires the generation of multiple high-resolution images, which can be a significant increase in computation time. Overall, we believe these to be exciting directions for future work.

## Impact Statement

This paper presents work whose goal is to advance the field of trustworthy machine learning. However, it is important to note that advances in image super-resolution technology can have significant societal impacts. For example, as much as our work enables the trustworthy use of super-resolution methods in crucial fields such as medical imaging, it could also be leveraged for harmful tasks such as reversing image censoring (e.g. censoring via blurred pixels). But we highlight that our approach's capability is generally constrained by the capabilities of the base super-resolution model, which we believe limits the negative impact of our work.

## Acknowledgements

This project is part of a global initiative funded by the International Development Research Centre (IDRC) on AI for Global Health, in collaboration with the UK International Development. GTG acknowledges partial funding from FAPERJ.

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

# A Proofs

*Proof of Theorem 2.1.* This proof is done via the standard conformal risk control argument [Angelopoulos et al., 2022a].

Consider the "lifted" threshold

$$t_\alpha^{(n+1)} = \sup \left\{ t \in \mathbb{R} \cup \{+\infty\} : \frac{1}{n+1} \sum_{i=1}^{n+1} \sup_{p;[\sigma(X_i)]_p \leq t} D_p\left(\mu(X_i), Y_i\right) \leq \alpha \right\},$$

which, as opposed to their "unlifted" counterpart $t_\alpha$, leverage the $(n+1)$-th sample as well and does not include the $3/(n+1)$ term.

Note that it is certainly the case that $t_\alpha^{(n+1)} \geq t_\alpha$. Moreover, note that the fidelity error is monotone with $t$, and so it suffices to show that the fidelity function with $t^{(n+1)}$ is upper bounded by $\alpha$.

Let $Z_*$ be the multiset of the samples $(X_i, Y_i)_{i=1}^n$ – i.e., a random variable representing the samples, but with their order discarded. Hence, upon conditioning on $Z_*$, all the randomness that remains is that of the order of the samples. It then follows:

$$\mathbb{E}\left[ \sup_{p;[\sigma(X_{n+1})]_p \leq t_\alpha^{(n+1)}} D_p(\mu(X_{n+1}), Y_{n+1}) \middle| Z_* \right] = \frac{1}{n+1} \sum_{i=1}^{n+1} \sup_{p;[\sigma(X_{n+1})]_p \leq t_\alpha^{(n+1)}} D_p(\mu(X_i), Y_i),$$

and by the definition of $t_\alpha^{(n+1)}$, this is upper bounded by $\alpha$. Thus

$$\mathbb{E}\left[ \sup_{p;[\sigma(X_{n+1})]_p \leq t_\alpha^{(n+1)}} D_p(\mu(X_{n+1}), Y_{n+1}) \right] = \mathbb{E}_{Z_*}\left[ \mathbb{E}\left[ \sup_{p;[\sigma(X_{n+1})]_p \leq t_\alpha^{(n+1)}} D_p(\mu(X_{n+1}), Y_{n+1})|Z_* \right] \right]$$

$$\leq \mathbb{E}_{Z_*}[\alpha] = \alpha,$$

which concludes the proof. □

*Proof of Proposition 3.1.* The PSNR we are bounding is given by

$$\mathbb{E}[\text{PSNR}\left(\mu(X_{n+1}), Y_{n+1}|M_\alpha(X_{n+1})\right)] := \mathbb{E}\left[ 10 \log_{10} \frac{(\max_{p \in M_\alpha(X_{n+1})}[Y_{n+1}]_p)^2}{|M_\alpha(X_{n+1})|^{-1} \sum_{p \in M_\alpha(X_{n+1})} ([\mu(X_{n+1})]_p - Y_p)^2} \right]$$

$$= 20\mathbb{E}\left[ \log_{10} \frac{\max_{p \in M_\alpha(X_{n+1})}[Y_{n+1}]_p}{\sqrt{|M_\alpha(X_{n+1})|^{-1} \sum_{p \in M_\alpha(X_{n+1})} ([\mu(X_{n+1})]_p - Y_p)^2}} \right].$$

Now, by Jensen's Inequality and standard properties of logarithms,

$$20\mathbb{E}\left[ \log_{10} \frac{\max_{p \in M_\alpha(X_{n+1})}[Y_{n+1}]_p}{\sqrt{|M_\alpha(X_{n+1})|^{-1} \sum_{p \in M_\alpha(X_{n+1})} ([\mu(X_{n+1})]_p - Y_p)^2}} \right]$$

$$= 20 \left( \mathbb{E}\left[ \log_{10} \max_{p \in M_\alpha(X_{n+1})}[Y_{n+1}]_p - \log_{10} \sqrt{|M_\alpha(X_{n+1})|^{-1} \sum_{p \in M_\alpha(X_{n+1})} ([\mu(X_{n+1})]_p - Y_p)^2} \right] \right)$$

$$= 20 \left( \mathbb{E}\left[ \log_{10} \max_{p \in M_\alpha(X_{n+1})}[Y_{n+1}]_p \right] - \mathbb{E}\left[ \log_{10} \sqrt{|M_\alpha(X_{n+1})|^{-1} \sum_{p \in M_\alpha(X_{n+1})} ([\mu(X_{n+1})]_p - Y_p)^2} \right] \right)$$

$$\geq 20 \left( \mathbb{E}\left[ \log_{10} \max_{p \in M_\alpha(X_{n+1})}[Y_{n+1}]_p \right] - \log_{10} \mathbb{E}\left[ \sqrt{|M_\alpha(X_{n+1})|^{-1} \sum_{p \in M_\alpha(X_{n+1})} ([\mu(X_{n+1})]_p - Y_p)^2} \right] \right);$$

And, because the RMSE is upper bounded by the maximum error, we get that

$$20\left(\mathbb{E}\left[\log_{10}\max_{p\in M_\alpha(X_{n+1})}[Y_{n+1}]_p\right]-\log_{10}\mathbb{E}\left[\sqrt{|M_\alpha(X_{n+1})|^{-1}\sum_{p\in M_\alpha(X_{n+1})}([\mu(X_{n+1})]_p-Y_p)^2}\right]\right)$$

$$\geq 20\left(\mathbb{E}\left[\log_{10}\max_{p\in M_\alpha(X_{n+1})}[Y_{n+1}]_p\right]-\log_{10}\mathbb{E}\left[\sup_{p\in M_\alpha(X_{n+1})}([\mu(X_{n+1})]_p-Y_p)\right]\right);$$

And, by Theorem 2.1,

$$20\left(\mathbb{E}\left[\log_{10}\max_{p\in M_\alpha(X_{n+1})}[Y_{n+1}]_p\right]-\log_{10}\mathbb{E}\left[\sup_{p\in M_\alpha(X_{n+1})}([\mu(X_{n+1})]_p-Y_p)\right]\right)$$

$$\geq 20\left(\mathbb{E}\left[\log_{10}\max_{p\in M_\alpha(X_{n+1})}[Y_{n+1}]_p\right]-\log_{10}\alpha\right)\geq -20\log_{10}\alpha,$$

where the last step holds as long as all pixel values are in $[0,1]$.

$\square$

*Proof of 3.2.* We effectively want to bound the supremum of the expected fidelity error as the leaked data is allowed to alter freely. For convenience, let $\sup_{\text{leaked}}$ denote the supremum over all possible values of the leaked samples $(X_i,Y_i)_{i=n_{\text{new}}+1}^n$ (and $\inf_{\text{leaked}}$ the corresponding infimum).

Note that the error function is decreasing on the selected parameter $t$ and continuous. Hence:

$$\sup_{\text{leaked}}\mathbb{E}\left[\sup_{p\in M_\alpha(X)}D_p(Y,\widehat{Y})\right]\leq\mathbb{E}\left[\sup_{\text{leaked}}\sup_{p\in M_\alpha(X)}D_p(Y,\widehat{Y})\right]=\mathbb{E}\left[\sup_{p;[\sigma(X)]_p\leq\sup_{\text{leaked}}t_\alpha}D_p(Y,\widehat{Y})\right],$$

and in turn

$$\sup_{\text{leaked}}t_\alpha=\sup_{\text{leaked}}\sup\left\{t\in\mathbb{R}\cup\{+\infty\}:\frac{1}{n+1}\sum_{i=1}^n\sup_{p;[\sigma(X_i)]_p\leq t}D_p(Y_i,\mu(X_i))+\frac{3}{n+1}\leq\alpha\right\}$$

$$\leq\sup\left\{t\in\mathbb{R}\cup\{+\infty\}:\inf_{\text{leaked}}\frac{1}{n+1}\sum_{i=1}^n\sup_{p;[\sigma(X_i)]_p\leq t}D_p(Y_i,\mu(X_i))+\frac{3}{n+1}\leq\alpha\right\}$$

$$=\sup\left\{t\in\mathbb{R}\cup\{+\infty\}:\frac{1}{n+1}\sum_{i=1}^{n_{\text{new}}}\sup_{p;[\sigma(X_i)]_p\leq t}D_p(Y_i,\mu(X_i))\right.$$

$$\left.+\inf_{\text{leaked}}\frac{1}{n+1}\sum_{i=n_{\text{new}}+1}^n\sup_{p;[\sigma(X_i)]_p\leq t}D_p(Y_i,\mu(X_i))+\frac{3}{n+1}\leq\alpha\right\}$$

$$=\sup\left\{t\in\mathbb{R}\cup\{+\infty\}:\frac{1}{n+1}\sum_{i=1}^{n_{\text{new}}}\sup_{p;[\sigma(X_i)]_p\leq t}D_p(Y_i,\mu(X_i))+\frac{3}{n+1}\leq\alpha\right\}$$

$$=\sup\left\{t\in\mathbb{R}\cup\{+\infty\}:\frac{1}{n_{\text{new}}+1}\sum_{i=1}^{n_{\text{new}}}\sup_{p;[\sigma(X_i)]_p\leq t}D_p(Y_i,\mu(X_i))+\frac{3}{n_{\text{new}}+1}\leq\alpha\cdot\frac{n+1}{n_{\text{new}}+1}\right\}.$$

Note that this corresponds to doing our calibration procedure only on the new data but with altered fidelity level $\alpha\cdot(n+1)/(n_{\text{new}}+1)=\alpha\cdot(n_{\text{new}}+n_{\text{leaked}}+1)/(n_{\text{new}}+1)$, and so, by the same arguments as in Theorem 2.1,

$$\mathbb{E}_{(X_i,Y_i)_{i=1}^{n+1}}\left[\sup_{p\in M_\alpha(X_{n+1})}D_p\left(\mu(X_{n+1}),Y_{n+1}\right)\right]\leq\alpha\cdot\frac{n_{\text{new}}+n_{\text{leaked}}+1}{n_{\text{new}}+1}.$$

$\square$

## B    On [Kutiel et al., 2023]

The method of [Kutiel et al., 2023] produces, given a low-resolution image $X$ in $[0,1]^{w \times h \times 3}$ and an initial estimate of uncertainty $\sigma$ in $\mathbb{R}_{\geq 0}^{kw \times kh}$, a predicted high-resolution image $\hat{Y}$ in $[0,1]^{kw \times kh \times 3}$ along with a continuous 'confidence mask' $M$ in $[0,1]^{kw \times kh}$, where higher values denote higher confidence at that region of the image. Ideally, this continuous mask would be such that it would satisfy an RCPS-like guarantee

$$\mathbb{P}\left[\mathbb{E}\left[\frac{1}{kw\,kh}\sum_{i,j}[M]_{i,j}\cdot\left|[\hat{Y}]_{i,j}-[Y]_{i,j}\right|\right]\leq\alpha\right]\geq 1-\delta, \tag{3}$$

for some chosen $\alpha$ and $\delta$. This goal is enunciated in Definition 2 in [Kutiel et al., 2023].[2] Their calibration procedure is then presented in Section 4.2 of their paper; it does the following:

1. For each $i = 1, \dots, n$, compute

$$M^{(\lambda)} := \min\left\{\frac{\lambda}{1-[\sigma]_{i,j}+\epsilon},1\right\}$$

$$\lambda_i := \max\left\{\lambda : \frac{1}{kw\,kh}\sum_{i,j}[M^{(\lambda)}]_{i,j}\cdot\left|[\hat{Y}]_{i,j}-[Y]_{i,j}\right|\leq\alpha\right\}$$

2. Compute $\lambda := \mathrm{Quantile}_{1-\delta}(\lambda_1,\dots,\lambda_n)$.

3. The resulting "calibrated" masks are produced by $M^{(\lambda)}$.

However, they do not prove that this procedure satisfies Equation 3, other than a passing mention at the end of their Section 4.2. Indeed, the guarantee actually does not hold: intuitively this should be fairly immediate:

1. They write[3] "Finally, $\lambda$ is taken to be the $1-\delta$ quantile of $\{\lambda_k\}_{k=1}^n$, i.e. the maximal value for which at least $\delta$ fraction of the calibration set satisfies condition (5). Thus, assuming the calibration and test sets are i.i.d samples from the same distribution, the calibrated mask is guaranteed to satisfy Definition 2." However, their condition (5) states that

$$\mathbb{E}\left[\frac{1}{kw\,kh}\sum_{i,j}[M]_{i,j}\cdot\left|[\hat{Y}]_{i,j}-[Y]_{i,j}\right|\right]\leq\alpha,$$

with an expectation – this expectation is fundamentally absent in the calibration procedure.

2. Additionally, the quantile taken is a simple empirical quantile. But the guarantee we want is that the populational risk is bounded. To satisfy this, one would need to slightly tweak the quantile; this is analogous to how in RCPSs you would apply a concentration inequality (e.g. Hoeffding).

Indeed, we provide here a counterexample, in which their procedure does not satisfy their stated guarantee.

---

[2]The one difference is the presence of the normalization $\frac{1}{kw\,kh}$, which is absent in their paper (they refer to a simple 1-norm). However, we found it to be necessary in order for their method to function, and conjecture that it was accidentally omitted in their work.

[3]Modulo ajustments to notation

*Example* B.1. Suppose that $\sigma$ is a deterministic 0 mask for all inputs. Then, to show that Equation 3 does not hold, we just need that

$$\mathbb{P}\left[\mathbb{E}\left[\frac{1}{kw\,kh}\sum_{i,j}[M]_{i,j}\cdot\left|[\hat{Y}]_{i,j}-[Y]_{i,j}\right|\right]\le\alpha\right]<1-\delta$$

$$\iff\mathbb{P}\left[\mathbb{E}\left[\frac{1}{kw\,kh}\sum_{i,j}[M]_{i,j}\cdot\left|[\hat{Y}]_{i,j}-[Y]_{i,j}\right|\right]>\alpha\right]>\delta$$

$$\iff\mathbb{P}\left[[M]_{i,j}\mathbb{E}\left[\frac{1}{kw\,kh}\sum_{i,j}\left|[\hat{Y}]_{i,j}-[Y]_{i,j}\right|\right]>\alpha\right]>\delta$$

$$\iff\mathbb{P}\left[\mathbb{E}\left[\frac{1}{kw\,kh}\sum_{i,j}\left|[\hat{Y}]_{i,j}-[Y]_{i,j}\right|\right]>\frac{\alpha}{[M]_{i,j}}\right]>\delta,$$

for some $\alpha$ and $\delta$. Further suppose that there is some nonzero probability $\tau$ that $\frac{1}{kw\,kh}\sum_{i,j}\left|[\hat{Y}]_{i,j}-[Y]_{i,j}\right|=0$ for all samples in the calibration set. Then, with this nonzero probability, $\lambda=1$ regardless of $\delta$, implying that $[M]_{i,j}=1/(1+\epsilon)$. By chosing $\delta=\tau/2$ and $\alpha\le\frac{1+\epsilon}{2}\mathbb{E}\left[\frac{1}{kw\,kh}\sum_{i,j}\left|[\hat{Y}]_{i,j}-[Y]_{i,j}\right|\right]$, we conclude.

## C  Pseudocodes

Concrete implementations of these algorithms can be found in our code.

---

**Algorithm 1** Conformal mask calibration (computation of $t_\alpha$) with dynamic programming

---

$T\leftarrow$ all unique values of $\sigma(X_1),\dots,\sigma(X_n)$
$T\leftarrow\mathrm{Sort}(T)$
$D\leftarrow$ values of $D_p$ each entry of $T$ $\qquad\qquad\qquad\triangleright$ can be computed jointly with the sorting
$I\leftarrow$ indices of the original images for each entry of $T$ $\triangleright$ can be computed jointly with the sorting
$R_i\leftarrow0$ $\qquad\qquad\qquad\qquad\qquad\qquad\qquad\qquad\triangleright$ risk so far on each observation
$R\leftarrow\frac{3}{n+1}$ $\qquad\qquad\qquad\qquad\qquad\qquad\qquad\qquad\triangleright$ total risk so far
$t_\star\leftarrow-\infty$
$l\leftarrow\mathrm{NA}$ $\qquad\qquad\qquad\qquad\qquad\qquad\qquad\triangleright$ last threshold seen so far
**for** $i=1,\dots,nd$ **do**
$\quad$**if** $T_i\ne l$ **then**
$\quad\quad$**if** $R\le\alpha$ **then**
$\quad\quad\quad t_\star\leftarrow T_i$
$\quad\quad$**end if**
$\quad$**end if**
$\quad l\leftarrow T_i$
$\quad r\leftarrow\max\{R_{I_i},D_i\}$
$\quad R\leftarrow R-\frac{1}{n+1}R_{I_i}+\frac{1}{n+1}r$
$\quad R_i\leftarrow r$
**end for**
**return** $t_\star$

---

**Algorithm 2** Conformal mask calibration (computation of $t_\alpha$) with a brute force search

---

$T \leftarrow$ all unique values of $\sigma(X_1), \ldots, \sigma(X_n)$
$t_\star \leftarrow -\infty$
**for** $t \in T$ **do**
    compute risk $R \leftarrow \frac{1}{n+1} \sum_{i=1}^n \sup_{p;[\sigma(X_i)]_p \leq t} D_p(Y_i, \mu(X_i)) + \frac{3}{n+1}$
    **if** $R \leq \alpha$ **then**
        $t_\star \leftarrow \max\{t_\star, t\}$
    **end if**
**end for**
**return** $t_\star$

---

## D   Results under Distribution-Shift

| Fidelity Level | Semantic $D_p$ | Non-semantic $D_p$ |
|---|---|---|
| | Avg. PSNR | Avg. PSNR |
| $\alpha = 0.075$ | $13.34 \pm 6.91$ | $13.14 \pm 6.55$ |
| $\alpha = 0.100$ | $13.28 \pm 7.30$ | $12.36 \pm 4.47$ |
| $\alpha = 0.125$ | $13.01 \pm 7.11$ | $12.30 \pm 3.81$ |
| W/o our method | $11.73 \pm 0.39$ | $11.74 \pm 0.39$ |

Table 2: **Quantitative evaluation of our methods under distribution-shift.** We evaluate the average PSNR across different fidelity levels ($\alpha$) for both semantic and non-semantic $D_p$, as calculated on Table 1. We utilized the calibration performed on the LIU4K dataset to calculate this metrics on the validation set of DIV2K [Agustsson and Timofte, 2017, Timofte et al., 2017] with 100 samples from a different distribution. Overall our method controls the PSNR point-wisely, but the intervals show that performs better on an in-distribution scenario.

## E   On Metric Comparisons with Baseline Methods

Direct metric comparisons with the baseline methods of [Angelopoulos et al., 2022b] and [Kutiel et al., 2023] present fundamental challenges due to output modality differences. These methods produce continuous uncertainty scores rather than binary masks, making objective metric comparisons infeasible without modification. While one could threshold their outputs to obtain binary masks, doing so fundamentally undermines their statistical guarantees, as these methods were not designed for binary mask production. Furthermore, the thresholding process itself introduces methodological complications: for [Angelopoulos et al., 2022b], a natural threshold at $\alpha$ exists, but for [Kutiel et al., 2023], threshold selection requires an additional calibration split, introducing another degree of freedom that complicates fair comparison.

Given these concerns, we argue that forcing these methods into a common evaluation framework would produce misleading comparisons that do not reflect their intended use cases. Instead, we present qualitative visualizations that respect each method's design principles while highlighting the distinct advantages of our approach–namely, direct binary mask generation with maintained statistical guarantees and substantially tighter confidence regions (mean mask size of $0.73 \pm 0.07$ for our method at $\alpha = 0.1$).

## F   Results on Image Colorization

Here we showcase a direct adaptation of our method for image colorization rather than super-resolution. The method works similarly, suggesting a broad applicability beyond our domain.

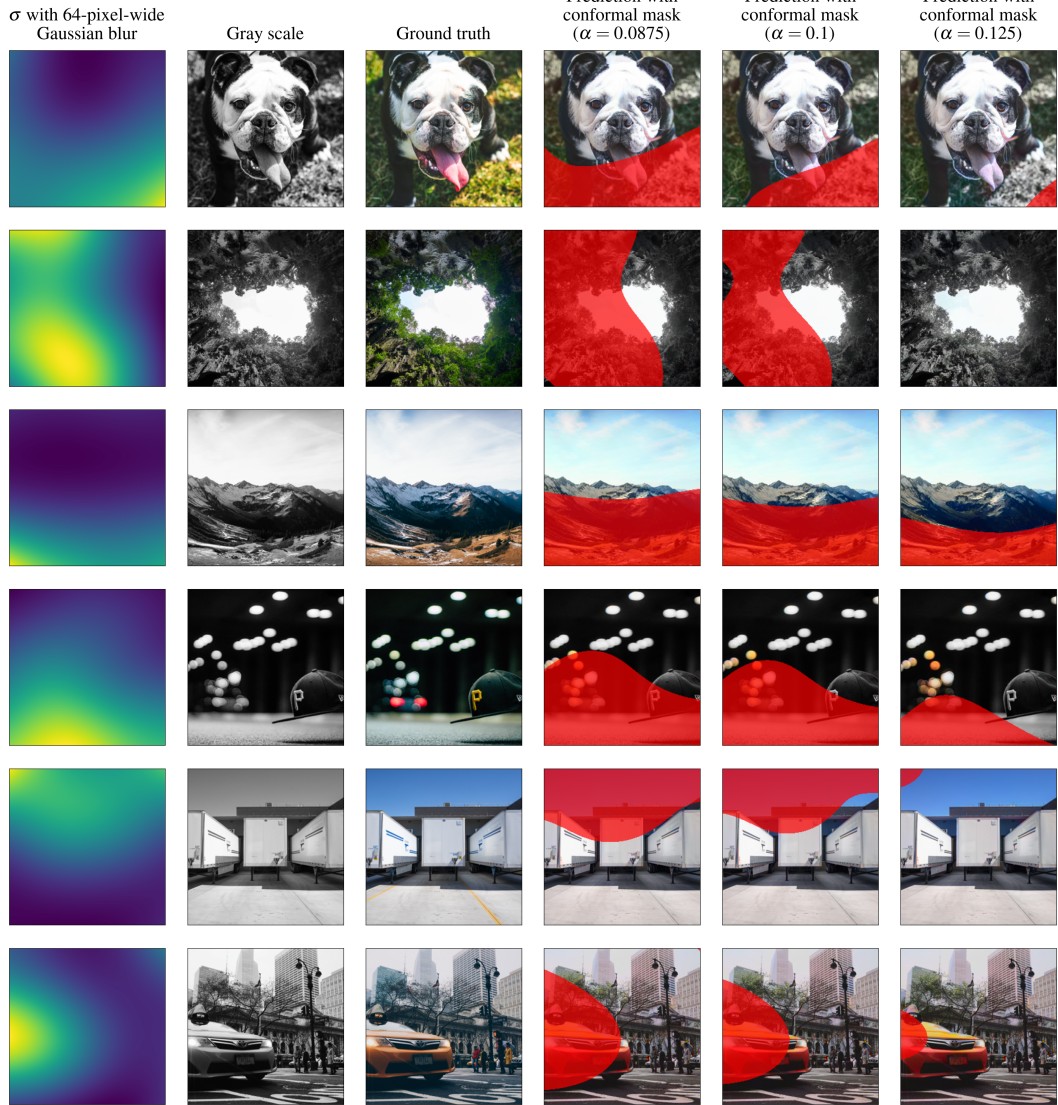

Figure 7: **Results of our method adapted for image colorization.** This figure presents a comparison of multiple color-restored images generated using the DDColor base model [Kang et al., 2023], alongside their corresponding conformal masks for $\alpha \in \{0.0875, 0.1, 0.125\}$ and non-semantic $D_p$. Our conformal masks accurately highlight regions where color predictions significantly deviate from the ground truth, effectively capturing uncertainty related to hue variations, saturation inconsistencies, and lighting discrepancies.This demonstrates the versatility of our theoretical framework, indicating it can be directly extended to other image restoration tasks beyond super-resolution.

