# OpenReview forum: "Image Super-Resolution with Guarantees via Conformalized Generative Models"
_NeurIPS.cc/2025/Conference — NeurIPS 2025 poster_

### Official Review · Reviewer_cVuS · 2025-06-19

**Clarity:** 2
**Significance:** 3
**Originality:** 2
**Rating:** 3
**Confidence:** 2

**Summary:**

This paper proposes an uncertainty quantification framework for generative image restoration models, leveraging conformal prediction to produce confidence masks that reflect the local reliability of generated pixels. The method is black-box compatible (requiring only outputs from the generative model), relies on a calibration dataset of high-resolution images, and supports flexible local similarity metrics for interpretability.

**Questions:**

1. How many samples are needed?

2. How does domain shift affect performance?

3. Does the choice of perceptual vs pixel-wise metric (e.g., LPIPS vs L1) affect the mask quality or calibration performance?

**Ethical Concerns:**

["NO or VERY MINOR ethics concerns only"]

**Final Justification:**

I thank the authors for their rebuttal. I have carefully reviewed the comments of the other reviewers. While the work is limited to a specific setting, there are still concerns about its generalization performance. Therefore, I will maintain my original score.

**Limitations:**

Yes

**Quality:**

2

**Strengths And Weaknesses:**

Strengths:

1. The method works on black-box models, which is practical for commercial or API-locked systems where internals are inaccessible.

2. The framework is grounded in conformal prediction, which gives valid coverage guarantees under minimal assumptions, and extends these ideas to the image domain with localization.

3. The idea of per-pixel confidence masks aligned with perceptual similarity metrics is appealing and potentially actionable for real users or downstream systems.

Weakness:

1. While the application to image restoration is interesting, the core method is a relatively straightforward application of standard conformal prediction, adapted to pixel-wise outputs. The “confidence mask” is essentially a localized conformal interval—novel in context, but not in construction.

2. The method assumes access to “a handful of unseen high-res images,” but doesn't quantify how much is sufficient or how sensitive the method is to domain shift between calibration and test inputs. This could be problematic for models trained on synthetic datasets and deployed on natural images.

3. There’s no empirical evidence (at least from intro/abstract) that the confidence masks are intuitive or actionable for users. User studies or visual comparison with ground-truth errors would help validate this interpretability claim.

---

> ### Author Rebuttal · Authors · 2025-07-30
>
> Thank you for your review. We have a few comments:
>
> **On the number of samples and domain shift:** much like usual conformal prediction methods, our calibration has fast $O(1/n)$ sample complexity, and so few samples are needed. We find that, in practice, as few as 50 samples are already enough for good guarantees; in the paper, our experiments use just 200. As for domain shift, it's straightforward to theoretically see that our method should be fairly robust: by a change-of-measure via total-variation (TV) distance, we have that for any distributions $P$ and $Q$, $\mathbb{E}\_Q[\text{risk}] \leq \mathbb{E}\_P[\text{risk}] + O(\mathbb{E}_X[TV(P|X, Q|X)])$, i.e., it's at most as bad as the expected total variation distance between the distributions conditional on the low-resolution image. We can add an experiment to empirically assess this in the final version.
>
> **Quantitative evidence on intuitiveness/actionability of our masks:** thank you for the suggestion. Note that we already have some comparisons with ground-truth images in Figures 1, 3 and 6; we'll add additional subplots with the pixel differences between the predicted and ground-truth images to facilitate visual inspection.
>
> **Perceptual vs. pixel-wise metric:** Changing the metric can lead to significant differences in mask quality. Moreover, the guarantee offered by our method changes, as it fundamentally involves the metric. Generally speaking, we find that non-semantic metrics lead to smaller confidence masks, likely due to our modelling of $\sigma$. As for calibration performance, both scale similarly with fast sample complexity of $O(1/n)$, and equal time complexity.
>
> Finally, note that our method is not quite standard conformal prediction, in which one would obtain predictive sets / conformal intervals (i.e., sets of predicted images), but rather an instance of conformal risk control, in which we have many more degrees of freedom on how to design our procedure. But yes, our main contribution is the use of such methods in our particular context.

---

> ### Comment · Reviewer_cVuS · 2025-08-05
> **Final Comment**
>
> I thank the authors for their rebuttal. I have carefully reviewed the comments of the other reviewers.The work is limited to a specific setting, and there are still concerns about its generalization performance. Therefore, I will maintain my original score.

---

### Official Review · Reviewer_E7RA · 2025-06-20

**Clarity:** 3
**Significance:** 3
**Originality:** 3
**Rating:** 4
**Confidence:** 3

**Summary:**

This paper introduces a conformal prediction-based method for uncertainty quantification in image super-resolution. The approach generates a binary "confidence mask" that identifies reliable regions in images upscaled by black-box generative models (e.g., diffusion models). The contributions include:

(1) A calibration procedure using easily obtained unlabeled data to quantify uncertainty working atop any black-box diffusion model.

(2) Theoretical guarantees for fidelity error control, PSNR bounds, and robustness to data leakage.

(3) The proposed approach not only works on super-resolution, but other tasks such as colorization.

**Questions:**

My main concerns are as follows:

1. How robust is the proposed approach given different σ (line108)? What if σ is poorly estimated or how to make sure a stable σ is obtained?

2. The evaluation of this paper is somewhat weak in demonstrating the accuracy of the proposed approach. Specifically, if my understanding is correct, the evaluation is mainly based on visualization comparisons in Fig.5 and Fig.6. These visualizations are kind of subjective and lack interpretability. No metric evaluation or human evaluation is involved to strengthen the effectiveness of the proposed method.

3.  The validation is simply conducted on the validation set of Liu4K, from the same source as the calibration data. The authors should consider testing on images from diverse sources to validate the robustness.

4. In the supp. material, the interpretability of the predicted mask is not that straightforward. Ideally, the red regions should indicate that the color is obviously different from the GT. But from my observation, the unmasked regions are also very different from the GT. I am kind of confused if the predicted results are accurate.

**Ethical Concerns:**

["NO or VERY MINOR ethics concerns only"]

**Final Justification:**

I have carefully read the author's rebuttal as well as other reviewers' comments.
I appreciate the theoretical analysis of this paper, which may introduce interpretability to the uncertainty of image super-resolution, a meaningful direction.
On the other hand, I am worried about the generalization ability, which seems hard to provide further verification in the submitted version.
I hereby tend to maintain my rating.

**Limitations:**

No.

The paper does not discuss the potential limitations and societal impact in any subsection of the paper. The authors should consider adding such a subsection in the revision.

**Quality:**

3

**Strengths And Weaknesses:**

Strengths:
+ The method provides provable statistical guarantees (Theorem 2.1, Propositions 3.1–3.2) for fidelity error, PSNR control, and data leakage robustness.
+ Binary masks offer clearer uncertainty visualization than pixel-wise intervals/continuous scores (Fig. 6).
+ Works with any black-box super-resolution model (including API-locked ones) and requires only unlabeled calibration data.

Weakness:
- The evaluation of this paper is somewhat weak in demonstrating the accuracy of the proposed approach. Specifically, if my understanding is correct, the evaluation is mainly based on visualization comparisons in Fig.5 and Fig.6. These visualizations are kind of subjective and lack interpretability. No metric evaluation or human evaluation is involved to strengthen the effectiveness of the proposed method.
- The validation is simply conducted on the validation set of Liu4K, from the same source as the calibration data. The authors should consider testing on images from diverse sources to validate the robustness.

---

> ### Author Rebuttal · Authors · 2025-07-30
>
> Thank you for your feedback. Below, we provide our responses to your comments.
>
> **On our evaluation:** we have three baselines: (i) not doing any uncertainty quantification, (ii) [Angelopoulos et al.] and (iii) [Kutiel et al.]. However, since the output of the latter two are of different modality from ours (i.e., they don't produce binary masks), we couldn't identify an objective metric with which to compare them. That said, following the suggestions of reviewer z7U3 we will add metric comparisons to thresholded versions of the two methods, even though by doing so they lose their statistical guarantees. In our preliminary experiments, our method seems to obtain substantially tighter confidence masks (for $\alpha$=0.1, our avg. mask size is 0.73 $\pm$ 0.07, binarized [Angelopoulos et al.] is 0.85 $\pm$ 0.18, and binarized [Kutiel et al.] is 0.88 $\pm$ 0.12).
>
> **Evaluation under distribution shift:** that is an interesting suggestion. We will add an experiment assessing this to the final version. Either way, it is already theoretically known that conformal prediction methods are generally robust to distribution shifts, so we expect this to behave favorably.
>
> **On the image colorization results in the suppl. mat.:** the perceived differences are due to a choice of $\alpha$ that is too high. Consider, for example, the image in the second row: since the predicted colored image is practically grayscale, the more saturated regions in the original have larger differences, and this is indeed what our method highlights. As the colors become less saturated, the prediction becomes more accurate, eventually being beneath the specified threshold given $\alpha$; this leads to them no longer being highlighted. We will add some comparisons with other values of $\alpha$ to the final version, which should clarify the accuracy of the method on all images.
>
> **Additional questions&points:**
>
> - Our guarantee that the fidelity error is controlled holds regardless of how well $\sigma$ is estimated; however, using a bad estimate may lead to masks that are larger than necessary. Generally speaking, though, the method seems reasonably robust to different $\sigma$. As for stabilization, note that the blur we apply on $\sigma$ effectively stabilizes it, making it so that small changes to the estimate are smoothened out and do not significantly affect the rest of the procedure.
> - We'll add a section discussing societal impact & limitations to the final version.
>
> [Angelopoulos et al.] "Image-to-Image Regression with Distribution-Free Uncertainty Quantification and Applications in Imaging" \
> [Kutiel et al.] "Conformal Prediction Masks: Visualizing Uncertainty in Medical Imaging"

---

> > ### Comment · Reviewer_E7RA · 2025-08-06
> >
> > I have carefully read the author's rebuttal as well as other reviewers' comments. I appreciate the theoretical analysis of this paper, which may introduce interpretability to the uncertainty of image super-resolution, which is a meaningful direction. On the other hand, I am worried about the generalization ability, which seems hard to provide further verification in the submitted version. I hereby tend to maintain my rating.

---

### Official Review · Reviewer_z7U3 · 2025-07-03

**Clarity:** 3
**Significance:** 2
**Originality:** 2
**Rating:** 4
**Confidence:** 5

**Summary:**

This paper addresses the need for interpretable uncertainty quantification in image restoration tasks, specifically focusing on image super-resolution using generative ML models. The authors propose a conformal prediction approach to generate a binary confidence mask that highlights regions where the generated image can be trusted. This method, adaptable to any black-box generative model, offers marginal coverage guarantees for a chosen local fidelity measure between the reconstructed and ground truth images, ensuring the expected maximum fidelity error across pixels is constrained by a user-specified level. They demonstrate that data leakage between calibration and training sets leads to a proportional increase in the fidelity level and establish that the method provides control over the PSNR conditioned on the mask. Experimental results, using the SinSR model on the Liu4K dataset, validate the method's effectiveness in quantifying and visualizing uncertainty in image recovery.

**Questions:**

* Please extend the related work and the comparison in the experimental section as mentioned in the weakness above.
* Please elaborate on the point of generating M images.
* The method seems general. Why restricting it application only to super-resolution?
* The use of semantic distance can be seen as providing per-pixel labels and then applying standard conformal risk control. Is this view correct?

**Ethical Concerns:**

["NO or VERY MINOR ethics concerns only"]

**Final Justification:**

Thank you for the rebuttal. Adding comparisons to more suitable baselines has partially addressed my concerns. I still believe the paper would be significantly improved by evaluating the proposed methods against other visualization techniques, even those lacking statistical guarantees. Nevertheless, I acknowledge the authors' efforts and the paper's contribution, and have raised my score accordingly.

**Limitations:**

See Questions and Weakness.

**Paper Formatting Concerns:**

No issues.

**Quality:**

2

**Strengths And Weaknesses:**

Strengths:
The paper is well-written and presents rigorous derivations for a method that visualizes uncertainty through a binary mask. This mask offers clear visualization and statistical guarantees regarding the recovery error. A significant strength of this approach is its reliance on conformal prediction, which inherently allows it to be applicable to any black-box generative model.

Weaknesses:
* The related work section could be more comprehensive. Given that the paper's main contributions are the interpretable visualization of uncertainty and clear statistical guarantees for recovery error, I expected a more extensive discussion of existing approaches for uncertainty quantification and visualization, both those that offer statistical guarantees and those that do not.

* The experimental evaluation appears limited. The authors do not compare their method to other competitive approaches for uncertainty visualization, even those that might not offer strict statistical guarantees (e.g. "Uncertainty quantification via neural posterior principal components"). Furthermore, the comparison to Angelopoulos's method is misleading; while that work did not explicitly aim to produce a mask, its pixel-wise prediction intervals could be used to generate a mask (as demonstrated in Figure 1 of "Image-to-Image Regression with Distribution-Free Uncertainty Quantification"). A more appropriate comparison would be against such a derived mask. Lastly, the comparison with Kutiel's method is unfair, as their continuous masks can be easily converted to binary ones via simple thresholding, an effort not undertaken by the authors.

* The method's requirement to produce M samples for generating each mask introduces additional computational complexity and increased runtime. This aspect should be thoroughly discussed in addition to clarifying which of the M images (if any single one) is ultimately presented to the user.

---

> ### Author Rebuttal · Authors · 2025-07-30
>
> We appreciate your review. Please check our responses below.
>
> **On the comparison to other methods:**
>
> - Regarding the display of [Angelopoulos et al.] in our Figure 6, the reviewer raises a good point. We will add the "uncertainty heatmaps" from their Figure 1.
> - On thresholding continuous masks: we agree that this is an interesting baseline, but note that in doing this we lose the underlying statistical guarantees, since the methods are not geared towards producing binary masks. There is also some nuance on how this threshold is to be chosen: for [Angelopoulos et al.] there is a natural choice of thresholding at the same $\alpha$ we use for our method, but for [Kutiel et al.] there is no such natural choice; we have to optimize for the threshold on an additional calibration split. Anyways, we'll add these comparisons. In preliminary experiments, we see that even when using their methods post-binarization, our method is substantially tighter (for $\alpha$=0.1, our avg. mask size is 0.73 $\pm$ 0.07, binarized [Angelopoulos et al.] is 0.85 $\pm$ 0.18, and binarized [Kutiel et al.] is 0.88 $\pm$ 0.12). We attribute this significant difference to the combination of our more direct statistical procedure with our modelling of $\sigma$.
> - We'll also add some discussion on methods without statistical guarantees and compare to the suggested work, thank you for the pointer.
>
> **On the M images:** this is indeed an additional cost, which we will try to make more explicit; however, that seems necessary to obtain a natural estimate of the uncertainty of the generative model. That said, note that this is trivially parallelized. As for which image to be presented to the user, the method and guarantee hold for any selection rule, as long as it is consistently applied. In all our experiments we simply use the first generated image.
>
> **Why focus on super-resolution?** The method is general, but for the sake of presentation and notation we felt it necessary to focus on one specific setting. Since this work was originally motivated by hallucinations in recent super-resolution methods (which are now even shipped on consumer devices!), we decided to go with that. Note, however, that we also show some results on image colorization in the supplementary material.
>
> **Re. semantic distance and "standard" conformal risk control:** the use of semantic distance can indeed be seen as providing per-pixel labels, but we do not believe that our method reduces to any particular proposal from the original Conformal Risk Control paper, besides following the general framework.
>
> [Angelopoulos et al.] "Image-to-Image Regression with Distribution-Free Uncertainty Quantification and Applications in Imaging" \
> [Kutiel et al.] "Conformal Prediction Masks: Visualizing Uncertainty in Medical Imaging"

---

### Official Review · Reviewer_g2HS · 2025-07-09

**Clarity:** 3
**Significance:** 3
**Originality:** 2
**Rating:** 4
**Confidence:** 4

**Summary:**

The authors work towards answering the interesting question of how to provide guarantees and interpretability for the output of diffusion-based generative models. Specifically, the authors propose a simple but effective method to quantify the uncertainty of images that are inpainted or super-resolved using diffusion models. The authors provide statistical guarantees their marginal conformal prediction will be bounded in expectation. The authors provide empirical evidence showing the effectiveness of their method and how tight the bounds are.

**Questions:**

## Questions

1. **Clarification of Proof Technique**
   Could the authors elaborate on the significance of their proof technique and explain how it differs from the original work by Anastasios et al?

2. **Tightness of the Bound**
   Could the authors comment on the tightness of their derived bound and discuss the main challenges involved in improving it?

**Ethical Concerns:**

["NO or VERY MINOR ethics concerns only"]

**Final Justification:**

The paper presents strong empirical results; however, its theoretical contribution lacks novelty and is largely a direct application of prior work outlined by the authors. The authors did not address concerns about the originality of their proposed theoretical component, despite it being a major part of the paper. Nevertheless, given the strength of the empirical findings, I lean toward accepting this work.

**Quality:**

3

**Strengths And Weaknesses:**

## Strengths

1. **Simplicity & Interpretability**
   The proposed algorithm is straightforward to implement and represents an important step toward greater interpretability in diffusion-based generative models. It also raises interesting philosophical questions about the trade off between fidelity and uncertainty (see Figure 2).

2. **Empirical Validation**
   The authors back up their theoretical claims with solid experimental evidence, demonstrating that the derived bounds are indeed appropriate in practice.

3. **Novel Application of Conformal Prediction**
   To the best of my knowledge, this work is the first to bring conformal prediction techniques into inverse problems specifically inpainting and super-resolution for diffusion-based generative models.


## Weaknesses

1. **Proof Novelty Is Unclear**
   Based on the Appendix, it’s difficult to distinguish the contributions of this proof from those in prior work by Anastasios Nikolas. The paper would benefit from a clearer discussion of what is fundamentally new.

2. **Loose Bounds**
   The bounds illustrated in Figure 4 appear rather loose, which makes it challenging to assess how informative they are in practice. Tighter bounds or additional commentary on their practical implications would strengthen the paper.

---

> ### Author Rebuttal · Authors · 2025-07-30
>
> Thank you for your comments. Please refer to our responses below.
>
> **On the novelty of the proof:** Theorem 2.1 indeed follows the arguments from [1], as we state in the proof. The novelty is mainly due to our choice of modelling and definition of the risk, which enables efficient computation and strong guarantees in our setting (there are also some minor differences regarding e.g. the direction of the continuity, but this is trivial). Theorem 2.1 aside, our Propositions 3.1 and 3.2 are entirely novel, and show that our method has desirable properties specific to our context, beyond those shown by [1].
>
> **On the tightness of our bounds:** there are two bounds in Figure 4.
>
> - **Re. bound under data leakage:** note that the bound is reasonably tight up until a certain point, corresponding to when our method becomes confident in the whole predicted images. That said, our bound is fairly pessimistic, since we consider that everything derived from the leaked samples can assume its worst-case value. While this is possible, the typical case is probably not as bad. To improve the bound, we would need to take into account training dynamics of specific model classes, which is best suited for future work.
> - **Re. bound on PSNR:** perhaps the lossiest step in the proof is a use of Jensen's inequality, which could be tightened by e.g. use of the generalized Jensen inequality. However, by doing so the bound will reference some unknown quantities of the underlying data distribution, making it less interpretable. Either way, in our view the main insight given by this result is that as we are more strict with our choice of $\alpha$, our PSNR increases accordingly, at a rate of $\Omega(-\log \alpha)$ (which seems natural for the PSNR).
>
> [1] "Conformal Risk Control", ICLR 2024

---

> > ### Comment · Reviewer_g2HS · 2025-08-06
> > **On the novelty of the proof**
> >
> > Could the authors please provide in detail what novelty in mathematical content that makes the proof interesting or significantly different from the previous work?

---

> > > ### Author Response · Authors · 2025-08-07
> > >
> > > As we mentioned in our rebuttal, the proof of Theorem 2.1 indeed follows directly from the arguments from the prior work of [1]. The novelty is our modelling of the image super-resolution problem, in particular our definition of the fidelity error (the sup of a customizable similarity metric over the predicted mask) as the object we want to control. This choice comes with the advantages of being fast to compute, easy to interpret and easily customizable; these are key advantages of our proposed method, and generally will not be true for other choices. It is also key for Proposition 3.1 to hold, where we show that our proposed procedure also controls notions of error usual to the computer vision literature (in particular, PSNR). Again, as noted, Propositions 3.1 and 3.2 are entirely new.
> > >
> > > We hope we managed to communicate the novelty of our work. If you have any questions or points you would like further clarification on, we'd be happy to answer.
> > >
> > > [1] "Conformal Risk Control", ICLR 2024

---

### Decision · Program_Chairs · 2025-09-17

**Decision:**

Accept (poster)

**Comment:**

This paper is about applying conformal prediction to super-resolution and other image generation models, which produces mask revealing uncertain pixel regions, and works with black box diffusion models with some theoretical guarantees.

Reviewers appreciated the simple method, empirical validation, excellent paper presentation, statisical guarantees, and connection with perceptual similarity metrics. Some weaknesses are the need for calibration data and not evaluating how much calibration data is needed.

I also note that this paper does not make a full comparison to the state of the art, which might not be strictly needed since the use of a uncertainty mask is novel and not commonly done in the literature. The reference list is on the short side and can be improved with discussion/comparison with state of the art uncertainty methods, for example in particular this one for super-resolution:
"Uncertainty Estimation for Super-Resolution using ESRGAN" at https://arxiv.org/abs/2412.15439

Considering the many strengths and little weaknesses, plus the wide applicability of the proposed method to image generation in diffusion models, in my opinion this paper can be accepted. I recommend the authors to consider the reviewer's comments in their final version.